# Observational Study of the Natural Growth History of Peripheral Small-Cell Lung Cancer on CT Imaging

**DOI:** 10.3390/diagnostics13152560

**Published:** 2023-08-01

**Authors:** Xu Jiang, Meng-Wen Liu, Xue Zhang, Ji-Yan Dong, Lei Miao, Zi-Han Sun, Shu-Shan Dong, Li Zhang, Lin Yang, Meng Li

**Affiliations:** 1Department of Diagnostic Radiology, National Cancer Center/National Clinical Research Center for Cancer/Cancer Hospital, Chinese Academy of Medical Sciences and Peking Union Medical College, Beijing 100021, China; jiangxu0824@163.com (X.J.); liumengwen2021@163.com (M.-W.L.); zhangxuebreezy@163.com (X.Z.); 18863608538@163.com (L.M.); zhangli_cicams@163.com (L.Z.); 2Department of Pathology, National Cancer Center/National Clinical Research Center for Cancer/Cancer Hospital, Chinese Academy of Medical Sciences and Peking Union Medical College, Beijing 100021, China; djy0823lucky@163.com (J.-Y.D.); zh_sun22@163.com (Z.-H.S.); 3Clinical Science, Philips Healthcare, Beijing 100600, China; mail_dong33@163.com

**Keywords:** small-cell lung cancer (SCLC), solid nodule, CT, volume-doubling time (VDT), time factor

## Abstract

Background: This study aimed to investigate the natural growth history of peripheral small-cell lung cancer (SCLC) using CT imaging. Methods: A retrospective study was conducted on 27 patients with peripheral SCLC who underwent at least two CT scans. Two methods were used: Method 1 involved direct measurement of nodule dimensions using a calliper, while Method 2 involved tumour lesion segmentation and voxel volume calculation using the “py-radiomics” package in Python. Agreement between the two methods was assessed using the intraclass correlation coefficient (ICC). Volume doubling time (VDT) and growth rate (GR) were used as evaluation indices for SCLC growth, and growth distribution based on GR and volume measurements were depicted. We collected potential factors related to imaging VDT and performed a differential analysis. Patients were classified into slow-growing and fast-growing groups based on a VDT cut-off point of 60 days, and univariate analysis was used to identify factors influencing VDT. Results: Median VDT calculated by the two methods were 61 days and 71 days, respectively, with strong agreement. All patients had continuously growing tumours, and none had tumours that decreased in size or remained unchanged. Eight patients showed possible growth patterns, with six possibly exhibiting exponential growth and two possibly showing Gompertzian growth. Tumours deeper in the lung grew faster than those adjacent to the pleura. Conclusions: Peripheral SCLC tumours grow rapidly and continuously without periods of nongrowth or regression. Tumours located deeper in the lung tend to grow faster, but further research is needed to confirm this finding.

## 1. Introduction

Lung cancer is the leading cause of cancer-related deaths worldwide, accounting for approximately 18.0% of all cancer deaths [1]. Based on pathological and histological characteristics, lung cancer is classified into small-cell lung cancer (SCLC) and non-small-cell lung cancer (NSCLC). SCLC, which accounts for approximately 15% of all lung cancer cases, is highly aggressive and frequently metastasizes to regional lymph nodes and distant organs, resulting in poor prognosis [2,3,4]. SCLC’s 5-year overall survival rate is only 7% [5]. However, early-stage SCLC patients (pathology T1~2N0M0) who undergo surgical treatment have a favourable 5-year overall survival rate (OS) of up to 50% [6,7,8]. Therefore, early detection and accurate diagnosis of early-stage SCLC and timely clinical intervention are critical for improving the patient’s survival rate.

Computed tomography (CT) is the optimal method for screening and diagnosing lung cancer [9]. With the widespread use of CT, detecting small, noncalcified, and uncertain peripheral pulmonary nodules has been increasingly recognized [9,10]. However, accurately diagnosing such nodules on imaging is challenging, as they are often small and difficult to biopsy. In clinical practice, CT follow-up observation of nodule changes is the most commonly adopted management strategy for peripheral pulmonary nodules [11]. Volume-doubling time (VDT) is an essential indicator used to evaluate the growth characteristics of lung cancer during clinical follow-up. It represents the time required for the lesion to double in size. VDT is an accurate parameter for determining the benign or malignant nature of pulmonary nodules and evaluating the invasiveness of malignant tumours [12]. It is also an independent prognostic factor for lung cancer patients [13,14]. A shorter VDT reflects stronger histological tumour invasiveness and indicates a poorer prognosis for lung cancer. The VDTs of clinically diagnosed lung cancer range from 20 to 360 days [15,16]. Numerous studies have detailed the growth rates of different lung adenocarcinoma subtypes, including solid, part-solid, and pure ground-glass opacity adenocarcinomas, each with different VDTs [17,18]. However, limited research has been conducted on the natural history of peripheral SCLC growth.

In previous studies, SCLC was mainly considered a central type of lung cancer [19]. However, recent studies have shown that the incidence of peripheral SCLC has been increasing, surpassing that of central SCLC [20,21,22]. Early peripheral SCLC often presents as small, solid nodules with uniform density and smooth edges on CT scans [23]. If an early diagnosis of peripheral SCLC cannot be made through qualitative determination from the initial CT scan and cannot be confirmed by needle biopsy, differences in growth characteristics between SCLC and other lung nodules can be used to determine its nature. Therefore, this study aims to investigate the natural growth history of peripheral SCLC using observation indicators such as the VDT and the growth curve to evaluate the growth rate of peripheral SCLC accurately.

## 2. Materials and Methods

### 2.1. Patients

This retrospective study was approved by the Research Ethics Committee of the National Cancer Hospital, Chinese Academy of Medical Sciences, and the patient’s informed consent requirement was waived. Eligible patients included those who underwent surgical resection for peripheral SCLC at the Cancer Hospital, Chinese Academy of Medical Sciences, between January 2015 and June 2022 and met the following inclusion criteria: (1) peripheral SCLC, defined as a lesion occurring below the bronchus to the segmental bronchus; (2) complete surgical resection with definite pathological diagnosis; and (3) at least two preoperative CT scans performed at intervals of at least 20 days. If there were two lesions, the larger lesion was selected for the study. Exclusion criteria were as follows: (1) incomplete clinical, radiological, or pathological data, (2) SCLC receiving neoadjuvant treatment during CT follow-up, and (3) poor image quality due to respiratory artefacts, bleeding during the biopsy, or other factors that could affect diameter measurement or volume contouring (Figure 1).

The images were transferred to the Picture Archiving and Communication System (PACS) for review. An experienced radiologist with at least two years of expertise in chest disease imaging reviewed the images without prior knowledge of the patient’s clinical information. In cases of disagreement, a final decision was made after consultation with a chief radiologist with more than 14 years of experience in chest imaging. Another experienced radiologist with expertise in chest disease imaging accessed the patients’ clinical records through the hospital’s PACS system and recorded relevant clinical information, including age, sex, history of previous tumours, family history of lung cancer, lesion location (upper and middle vs. lower), smoking history, smoking index, tumour-node metastasis (TNM) staging, and several preoperative chest CT scans. In cases where the patient had a history of malignant tumours, simultaneous SCLC was noted. TNM staging was determined according to the 8th edition of the American Joint Committee on Cancer (AJCC) staging manual [24].

### 2.2. Image Acquisition

All patients underwent CT scanning using the following equipment: Optima CT660, Revolution CT, Bright-Speed CT, Toshiba Aquilion 64-slice spiral CT and Discovery CT750 (GE Medical System, Milwaukee, WI, The Unites States of America). Before CT scanning, patients received respiratory training. Patients were placed supine with either the head or feet tilted forward and instructed to hold their breath during scanning after a deep inhalation. CT examination was performed with a tube voltage of 120 kVp, a tube current of 200–350 mAs, and a slice thickness of 5 mm. Partial reconstruction was performed with a slice thickness ranging from 0.75 mm to 1.25 mm. If contrast-enhanced scanning was necessary, iopromide injection (iodine concentration of 300 mg/mL) was used with a dose of 80–90 mL and a flow rate of 2.5–3.0 mL/s. Standard and high-resolution algorithms were used for image and parallel multiplanar reconstruction. The lung window (window width 1600 HU, window level −400 HU) and mediastinal window (window width 400 HU, window level 40 HU) were selected for image observation.

### 2.3. Image Analysis and VDT Calculation

Two methods were used for measurements. For Method 1, a radiology graduate student measured directly on the DICOM images on the PACS workstation using a calliper. The maximum axial diameter (X), corresponding perpendicular diameter (Y), and head-to-tail diameter (Z) of the nodule were measured, and MPR reconstruction was performed. The X, Y, and Z diameters were then used in the following formula to calculate the VDT: VDT = [ln2 × ∆T]/[ln((MaxDiamXY2 × PerpDiamXY2 × MaxDiamZ2)/(MaxDiamXY1 × PerpDiamXY1 × MaxDiamZ1))]; MaxDiamXY1-2 = maximum nodule diameter in the X/Y axis, PerpDiamXY1-2 = perpendicular diameter to the maximum diameter in the X/Y axis, MaxDiamZ1-2 = maximum diameter in the Z axis, ∆T = time (in days) between the two scans, and ln = natural logarithm [10].

For Method 2, DICOM images output from the PACS archive were anonymously processed. The images were resampled to a uniform voxel size of 0.5 mm × 0.5 mm × 0.5 mm to eliminate differences between images with different layer thicknesses. The tumour lesion was contoured layer by layer along the pulmonary window using ITK-SNAP software (Version 3.8.0, 12 June 2019), with attempts to exclude adjacent normal tissues such as vessels and bronchi and a collapsed lung. The contouring was performed by a radiologist specializing in chest imaging, and the results were confirmed by a chest imaging specialist (Figure 2). The voxel volume feature was extracted using the “pyradiomics” package in Python and multiplied by the volume of a single voxel to obtain the volume value for two or more scans of the same patient. The VDT was then calculated using the modified Schwartz equation (VDT = t × log2/log(V2/V1)), where t is the scan interval time, V2 is the volume of the second scan, and V1 is the volume of the first scan. If there were three or more CT scans, the formula was used with the first and last scans.

We used VDT and GR (growth rate, GR = 1/VDT) as evaluation indices for SCLC growth [25]. Furthermore, we depicted the distribution of SCLC growth based on GR and the growth curve based on volume measurements from three or more follow-up CT scans. To minimize measurement discrepancies, we endeavoured to compare and analyse images with the most similar scan thickness before and after the measurement. We utilized the thinnest possible slice images whenever feasible.

### 2.4. Statistical Analysis

We assessed the normality of parameter data using t tests and Pearson correlation analysis. Nonnormal parameter data were analysed using Mann–Whitney U tests and Spearman correlation analysis. We evaluated the consistency of measurements using the intraclass correlation coefficient (ICC). Additionally, we used a cut-off point of VDT = 60 days to classify patients into slow-growing and fast-growing groups [26]. Univariate analysis was conducted to identify factors influencing VDT, with *p* < 0.05 indicating statistical significance. Statistical analyses were performed using R version 4.1.0 and SPSS 26.0.

## 3. Results

In total, 27 patients met the inclusion criteria (Figure 1). A total of 63 CT scans were performed, with 19 patients undergoing two scans, seven patients undergoing three scans, and one patient undergoing four scans. Among them, 27 scans (43%) were plain, and 36 scans (57%) were enhanced. Thin-layer scans accounted for 76%, while 15 (24%) were 5 mm thick.

Table 1 and Table 2 presented the clinical data and tumour characteristics of the 27 patients. All patients underwent thoracoscopic surgery with systematic lymph node dissection and had no distant metastasis before surgery. One patient had bifocal SCLC, while the remaining patients had solitary SCLC. The pathological staging is presented in Table 3. The majority of cases were early-stage SCLC, with T1N0 and T2N0 accounting for 51.8% (*n* = 14) of the cases, followed by T1N2 (18.5%, *n* = 5) and T1N1 (11.1%, *n* = 3). T2N1 and T2N2 each constituted 7.4% (*n* = 2) and 7.4% (*n* = 2) of the cases, respectively. T3N0 was the least common stage, observed in only 3.7% (*n* = 1). Among the 25 patients who underwent immunohistochemistry testing, the median value of Ki-67 was 70% (IQR: 60–80%). Pathological analysis showed STAS in 8 patients (29.6%). As of December 2022, prognostic information was available for 20 patients, while seven patients were lost to follow-up (Appendix A). In addition, we observed that 4 cases were detected during follow-up CT scans between two consecutive scans, suggesting a ‘de Novo appearance of the small-cell lung cancer lesions (Table 4).

The two methods used for calculating the median VDT produced values of 61 days (IQR: 51–104 days) and 71 days (IQR: 49–103 days), respectively, with strong agreement (α = 0.90). The corresponding median GR values for the two VDT values were 0.0164 ± 0.0081 and 0.0152 ± 0.0075, respectively, with strong agreement (α = 0.83). Table 1 and Table 2 summarize various patient and tumour characteristics that may be associated with VDT and GR. The results showed that the second method yielded more SCLC patients with pleural attachment growth than those with lung growth, and the difference was statistically significant (*p* = 0.04).

There were no significant differences in VDT based on nonparametric data, including sex, smoking history, previous cancer history, family history of lung cancer, location (upper and middle lobes vs. lower lobes), shape, margin (lobulated vs. spiculated), and lung background (emphysema or interstitial pneumonia changes) (*p* > 0.05) (Table 1 and Table 2). There were no clear correlations between VDT and parametric data, including smoking index (Spearman’s rank rho VDTM-interval days = 0.24, *p* = 0.22; Spearman’s rank rho VDTR-interval days = 0.12, *p* = 0.54), baseline maximum diameter (Spearman’s rank rho VDTM-interval days = 0.05, *p* = 0.80; Spearman’s rank rho VDTR-interval days = 0.15, *p* = 0.47), and baseline volume (Spearman’s rank rho VDTM-interval days = −0.11, *p* = 0.57; Spearman’s rank rho VDTR-interval days = 0.19, *p* = 0.33) (*p* > 0.05). There was no correlation between the two methods of measuring VDT and age (Spearman’s rank rho VDTM-interval days = 0.13, *p* = 0.51; Spearman’s rank rho VDTR-interval days = 0.09, *p* = 0.67) or scan interval time (Spearman’s rank rho VDTM-interval days = 0.04, *p* = 0.85; Spearman’s rank rho VDTR-interval days = −0.09, *p* = 0.66). In addition, we used a VDT of 60 days to divide the peripheral SCLC patients into two groups: a slow-growing group with a VDT greater than or equal to the median value and a fast-growing group with a VDT less than the median value. Based on the VDT distribution of peripheral SCLC, there were ten patients in the fast-growing group and 17 in the slow-growing group. We attempted to perform univariate analysis, but none of the characteristics in Appendix B showed statistical significance (*p* > 0.05). Figure 3 illustrates the growth distribution of SCLC. Fourteen patients (52%) had a VDT greater than 67 days. Figure 4 depicts the growth curves of SCLC in 8 patients who underwent three or more CT follow-up examinations.

## 4. Discussion

According to our research results, the VDT of peripheral SCLC using the first method was 61 days (IQR: 51–104 days), and that using the second method was 71 days (IQR: 49–103 days). Generally, the VDT of lung cancer is less than 400 days [27], consistent with our findings. There have been many studies on the VDT of SCLC. Friberg and Mattson [28] reported a median VDT of 64 days in 23 cases of SCLC. Sone et al. [29] reported a median VDT of 54 days in 5 cases of SCLC. Rebecca et al. [30] reported a median VDT of 58 days in 3 cases of SCLC (2 central type cases and one peripheral type case). However, these studies did not differentiate between peripheral and central-type SCLC cases. Central-type SCLC is often difficult to distinguish from lymph nodes in the mediastinum or hilum, which may lead to errors in measuring the VDT. In our study, SCLC lesions appeared as soft tissue density shadows on CT images, with clear edges and a significant density difference from the surrounding tissue, facilitating various measurements and volume analysis. To our knowledge, there are few studies on the VDT of peripheral SCLC. Zhang et al. [31] reported a range of VDTs of 37–215 days with a median of 78.5 days in 10 cases of peripheral SCLC, which is consistent with our findings.

Additionally, in our study, we observed four cases of “de novo” small-cell lung cancer, which refers to the appearance of a new SCLC lesion on a CT scan after a previous scan showed no evidence of a lesion. The time intervals between scans were 87 days, 87 days, 90 days, and 138 days. Therefore, we must be vigilant about the possibility of SCLC when a new peripheral solid nodule appears within a short period of time. These observations suggest that SCLC has a short volume doubling time, a rapid growth rate, high malignancy, and strong invasiveness. Our study results and previous research [28,31], we suggest that patients suspected of having peripheral SCLC should undergo follow-up CT examination no later than two months (60 days).

In our study, two methods were used to measure VDT. Traditionally, tumour growth evaluation is accomplished through manual two-dimensional (2D) or three-dimensional (3D) linear measurement on chest X-ray or CT images, followed by estimation of tumour volume using a standard formula [32], similar to the first method used in our study. In addition, our second method employed image segmentation by radiologists using ITK-SNAP software to obtain a volume model. The texture features were extracted using Py-Radiomics to obtain volume features (Voxel Volume containing all voxels of the model). The volume of the model was obtained by summing the volumes of all voxels. Then VDT was calculated using the Schwartz equation, similar to the calculation model principles in previous studies [33]. The consistency analysis of the two methods showed strong consistency, which confirmed that our results were close to the true values. Since the second method accurately captured the actual nodule volume represented by the sum of all voxels in the segmentation model, we consider it more accurate.

We investigated the clinical, imaging, and pathological patient characteristics that may be related to VDT. In 27 patients, SCLC lesions located closer to the pleura grew faster than those located farther away (*p* = 0.04), which we speculated may be due to the proximity of the tumour to the heart and the increased blood supply. In studies of NSCLC, the growth rate was not associated with patient age or scan interval time [16,34], and our study showed the same result for SCLC. In addition, we divided the patients into slower- and faster-growth groups based on a VDT of 60 days. According to the VDT distribution of SCLC, were ten patients in the faster growth group and 17 patients in the slower growth group. We attempted single-factor logistic regression analysis but found no statistically significant features (*p* > 0.05), which may be related to the small sample size and warrants further exploration in larger studies.

Figure 3 shows the distribution of growth rates in SCLC patients, with approximately 14 patients (52%) having a VDT greater than 60 days. None of the patients had a VDT ≤ 0, indicating that SCLC lesions were in a state of continuous growth and had not yet exhibited any nongrowth or shrinkage. If there is no growth or shrinkage in incidental nodules during follow-up CT examinations, a diagnosis of SCLC should not be considered. Figure 4 shows eight cases of SCLC displaying various growth patterns. Previous studies have shown that SCLC typically presents as a solid soft tissue density shadow with a high cancer cell density and unlimited proliferation [35,36], with an increase in the number of cells, usually translating into an increase in volume [17]. In this study, six patients (75%) exhibited slightly slower growth when the tumour volume was small, which then accelerated with increasing volume and showed stability on a logarithmic scale, consistent with an exponential growth model [17], especially during the follow-up of the only lesion with four CT examinations. In addition, two patients (25%) exhibited faster growth when the tumour volume was small, which then slowed down with increasing volume, which may be consistent with the Gompertzian growth model [25,37], where cancer cells initially grow exponentially, but the growth rate gradually slows down as the tumour volume increases. However, this is affected by factors such as the short observation time, the small number of CT examinations, and the small sample size; therefore, further research is needed. In summary, SCLC growth exhibits heterogeneity.

Our study has several limitations. First, it was a retrospective study and may have been subject to selection bias, as most patients selected only had CT images acquired at two-time points. Second, the relatively small sample size is due to the rarity of untreated follow-up cases in SCLC patients. Most of these patients are incidentally discovered, asymptomatic, and have small lesions at the time of detection, making them eligible for follow-up. Therefore, considering the characteristics of this specific patient population, the inclusion of 27 cases already represents a considerable number for this type of SCLC patient. Third, contrast agents can also affect nodule size [38], but many studies on VDT, including ours, have not controlled for this factor [39,40].

## 5. Conclusions

In conclusion, peripheral SCLCs exhibit rapid growth rates, and our study indicates that their growth process does not involve periods of growth cessation or regression. It is crucial to note that peripheral nodular lesions with a doubling time, like in this study, should be treated cautiously, as they may indicate lung cancer and SCLC. Moreover, we found that SCLC tumours within the lung grow faster than those adjacent to the pleura. However, our sample size was limited, and larger studies are required to validate this observation further.

## Figures and Tables

**Figure 1 diagnostics-13-02560-f001:**
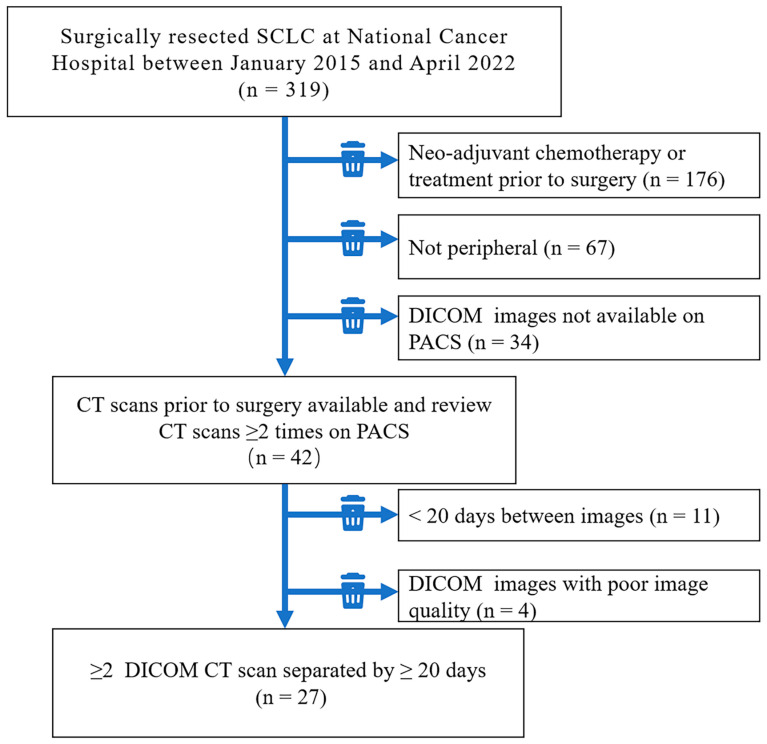
Flowchart depicting the patient selection process.

**Figure 2 diagnostics-13-02560-f002:**
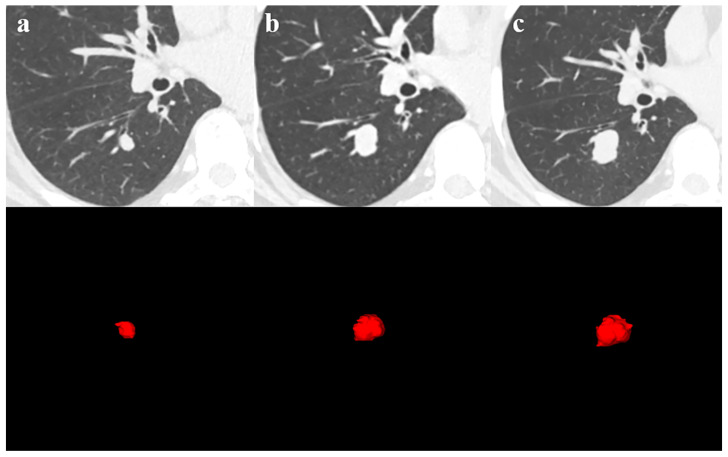
69-year-old man with SCLC. (**a**) The first CT scan, 20 February 2021, 630.5 mm^3^; (**b**) The second CT scan, 23 July 2021, 2490.9 mm^3^; (**c**) The third CT scan, 26 August 2021, 3895.75 mm^3^.

**Figure 3 diagnostics-13-02560-f003:**
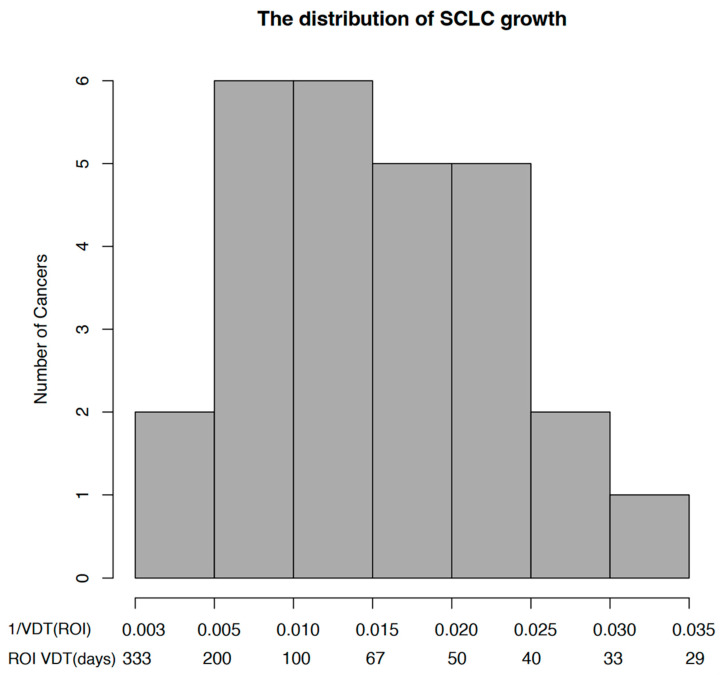
Distribution of growth rates of 27 surgically resected SCLC. The x-axis is scaled using the growth rate (GR, 1/VDT). The corresponding VDT is labelled below each GR.

**Figure 4 diagnostics-13-02560-f004:**
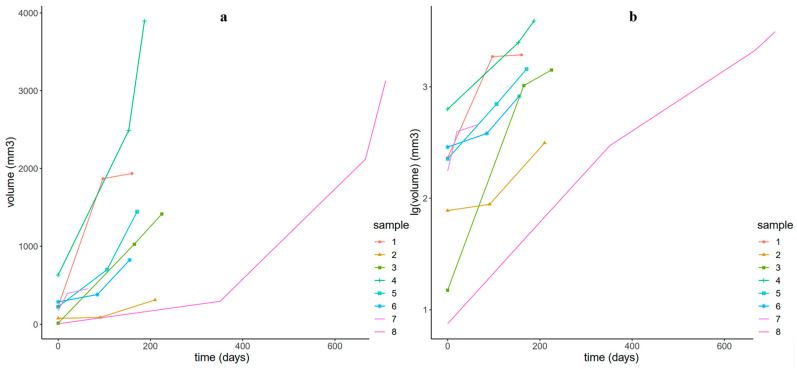
8 Surgically resected SCLC growth curves with three or more comparable CT scans. (**a**) A linear scale (**b**) A log scale. On the log scale, most growth curves are approximately linear, consistent with exponential growth.

**Table 1 diagnostics-13-02560-t001:** Patient characteristics and volume-doubling time.

Clinical Characteristic	Total	VDT (Days)		*p* Value (Manual)	*p* Value (ROI)	GR	
	N = 27 (100%)	Manual	ROI			Manual	ROI
		61 (51~104)	71 (49~103)			0.0164 ± 0.0081	0.0152 ± 0.0075
Age	64.8 ± 6.4						
Sex							
Male	25 (92.6%)	61 (50~105)	71 (48~104)	1.00	0.89	0.0164 (0.0095~0.0200)	0.0141 (0.0097~0.0206)
Female	2 (7.4%)	63 (60~66)	67 (65~69)			0.0161 (0.0153~0.0169)	0.0150 (0.0146~0.0155)
Smoking history							
Never	6 (22.2%)	59 (55~72)	67 (62~88)	0.93	0.71	0.0171 (0.0140~0.0183)	0.0150 (0.0115~0.0162)
Current or former	21 (77.8%)	63 (50~105)	71 (47~104)			0.0160 (0.0095~0.0200)	0.0141 (0.0097~0.0215)
Smoking index	600 (150~900)						
Tumour history							
Yes	10 (37%)	59 (48~92)	66 (48~104)	0.68	0.86	0.0169 (0.0114~0.0207)	0.0153 (0.0096~0.0209)
No	17 (63%)	68 (54~105)	71 (50~94)			0.0146 (0.0095~0.0187)	0.0141 (0.0107~0.0202)
Family history of Lung cancer							
Yes	3 (11.1%)	69 (68~89)	92 (81~113)	0.31	0.35	0.0146 (0.0119~0.0146)	0.0109 (0.0092~0.0125)
No	24 (88.9%)	59 (49~103)	63 (48~102)			0.0169 (0.0097~0.0204)	0.0158 (0.0098~0.0209)
Interval days	58 (30~169)						

VDT, volume-doubling time; GR, growth rate.

**Table 2 diagnostics-13-02560-t002:** Tumour characteristics and volume-doubling time.

Routine Imaging Characteristic	Total	VDT (Days)		*p* Value (Manual)	*p* Value (ROI)	GR	
	N = 27 (100%)	Manual	ROI			Manual	ROI
		61 (51~104)	71 (49~103)			0.0164 ± 0.0081	0.0152 ± 0.0075
Large diameter (mm)	19.2 (8.6~29.5)						
Size (mm)	17.9 (7.6~24.8)						
Volume (mm^3^)	2283 (202.9~6033.1)						
Site in the lung							
Upper and middle	15 (55.6%)	68 (47~93)	64 (47~93)	0.72	0.37	0.0146 (0.0109~0.0213)	0.0204 (0.0167~0.0222)
Lower	12 (44.4%)	61 (54~106)	72 (51~140)			0.0165 (0.0094~0.0185)	0.0138 (0.0072~0.0194)
Shape							
Regular	8 (29.6%)	61 (56~109)	63 (49~81)	0.37	0.68	0.0165 (0.0093~0.0179)	0.0158 (0.0126~0.0205)
Irregular	19 (70.4%)	55 (45~94)	67 (47~103)			0.0160 (0.0114~0.0219)	0.0123 (0.0097~0.0188)
Spiculated							
Yes	3 (11.1%)	207 (126~210)	237 (138~258)	0.24	0.31	0.0048 (0.0047~0.0137)	0.0042 (0.0039~0.0149)
No	24 (88.9%)	61 (52~89)	67 (49~96)			0.0165 (0.0114~0.0192)	0.0149 (0.0104~0.0203)
Lobulation							
Yes	16 (59.3%)	62 (46~106)	76 (48~111)	0.72	0.72	0.0162 (0.0094~0.0216)	0.0132 (0.0091~0.0209)
No	11 (40.7%)	61 (54~80)	64 (51~88)			0.0164 (0.0126~0.0186)	0.0157 (0.0117~0.0197)
Pleural retraction							
Yes	1 (3.7%)	119	158	0.30	0.30	0.0084	0.0063
No	26 (96.3%)	67 (49~100)	61 (51~97)			0.0165 (0.0103~0.0198)	0.0149 (0.0100~0.0205)
Location							
Pleural-attached	6 (22.2%)	50 (41~102)	130 (96~160)	0.44	0.04 *	0.0201 (0.0110~0.0247)	0.0081 (0.0062~0.0104)
Purely intraparenchymal	21 (77.8%)	63 (54~102)	63 (48~81)			0.0160 (0.0098~0.0185)	0.0159 (0.0123~0.0206)
Peripheral emphysema							
Yes	13 (48.1%)	54 (47~84)	61 (45~102)	0.26	0.2	0.0187 (0.0119~0.0213)	0.0164 (0.0098~0.0223)
No	14 (51.9%)	65 (58~107)	77 (62~101)			0.0153 (0.0093~0.0171)	0.0130 (0.0099~0.0162)
Interstitial pneumonia							
Yes	4 (14.8%)	54 (42~99)	106 (50~190)	0.67	0.53	0.0189 (0.0135~0.0247)	0.0127 (0.0056~0.0199)
No	23 (85.1%)	63 (53~104)	71 (49~98)			0.0160 (0.0096~0.0189)	0.0141 (0.0102~0.0204)

* Significant difference (*p* < 0.05). GR, growth rate; TNM, tumour, nodes, metastasis; VDT, volume-doubling time.

**Table 3 diagnostics-13-02560-t003:** Pathology stage.

TNM Stage	Total
	N = 27 (100%)
T1N0	10 (37.0%)
T2N0	4 (14.8%)
T3N0	1 (3.7%)
T1N1	3 (11.1%)
T1N2	5 (18.5%)
T2N1	2 (7.4%)
T2N2	2 (7.4%)

**Table 4 diagnostics-13-02560-t004:** 4 Cases were detected during follow-up CT scans between two consecutive scans.

No.	CT Scan Time for Newly Emerging Nodules			The Volume of Newly Emerging Nodules (mm^3^)
	Last Follow-Up	Newly Emerging Nodule	Interval Days	
1	12 August 2020	7 November 2020	87	230.5
2	8 November 2017	26 March 2018	138	47,635.4
3	25 October 2019	20 January 2020	87	15
4	24 December 2014	24 March 2015	90	162.5

## Data Availability

The data presented in this study are available on request from the corresponding author.

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
