# Peer review of "Observational Study of the Natural Growth History of Peripheral Small-Cell Lung Cancer on CT Imaging"

_diagnostics, 2023, doi:10.3390/diagnostics13152560_

Round 1

Reviewer 1 Report

This study has a significant impact on determining the rate of growth of peripheral small cell lung cancer to differentiate between fast growing and slow growing nodules based on volume doubling time parameter (VDT).

The technical works are moderate but the statistical analysis are extensive.

Fig 2 can be explained further in terms of the nature of the patient images as well as the dates of the scanning seem confusing. The date on (a) is later than (b)?

The results have been clearly outlined in Table 1-5. However some of the parameters have not been explained in Table 4 (pT pN. and so on)

Fig 3 could be well comprehended. Further elaboration is needed. How was the GR in x scale are selected and what is number of cancer?

A specific section on conclusion can be added.

Reviewer 2 Report

This is an important manuscript on the growth history of peripheral small-cell lung cancer (SCLC). More specifically in this retrospective study the authors aim to use CT scan images to detect doubling time in SCLC.

The population is carefully selected and certain inclusion criteria have been applied. Image processing was performed by expert radiologists and by the algorithm that the authors applied. In the results section the authors have investigated different characteristics that a nodule may have and their possible relationship with the doubling time of the cancer.

The study has important findings and below there are a few remarks that can be considered:
- Please redesign Fig.1 to improve demonstration.
- Please redesign Fig.3 to improve demonstration.
- Survival information as a table should be avoided.
- Please add some more text to explain Fig.4, which is not clearly presented.
- In Line 260 it is stated that "smoking is not correlated with doubling time". Probably, because of the innate potential of the lung tumor itself. Previous exposure has led to carcinogenesis. I feel the sentence can be removed to empower your other findings. Or add something (e.g. instead of other causes of nodular lesions that can be affected by smoking .....)
- It seems that based on your important findings, in the conclusions you could also state that "Peripheral nodular lesions with a doubling time like in this study, should be treated with caution, as they may be lung cancer, and even SCLC." (this is an example suggestion).
